# A Methodological Framework for Assessing the Benefit of SARS-CoV-2 Vaccination following Previous Infection: Case Study of Five- to Eleven-Year-Olds

**DOI:** 10.3390/vaccines11050988

**Published:** 2023-05-16

**Authors:** Christina Pagel, Harrison Wilde, Christopher Tomlinson, Bilal Mateen, Katherine Brown

**Affiliations:** 1Clinical Operational Research Unit, Department of Mathematics, University College London (UCL), London WC1E 6BT, UK; 2Department of Statistics, University of Warwick, Coventry CV4 7AL, UK; 3UCL Institute of Health Informatics, University College London (UCL), London NW1 2DA, UK; 4UK Research and Innovation Centre for Doctoral Training in AI-enabled Healthcare Systems, University College London (UCL), London WC1E 6BT, UK; 5University College London Hospitals Biomedical Research Centre, University College London (UCL), London W1T 7DN, UK; 6Wellcome Trust, London NW1 2BE, UK; 7Biomedical Research Centre, Great Ormond Street Hospital for Children, London WC1N 3JH, UK; 8Institute of Cardiovascular Science, University College London (UCL), London WC1E 6DD, UK

**Keywords:** mathematical modelling, paediatric vaccines, health policy, COVID-19

## Abstract

Vaccination rates against SARS-CoV-2 in children aged five to eleven years remain low in many countries. The current benefit of vaccination in this age group has been questioned given that the large majority of children have now experienced at least one SARS-CoV-2 infection. However, protection from infection, vaccination or both wanes over time. National decisions on offering vaccines to this age group have tended to be made without considering time since infection. There is an urgent need to evaluate the additional benefits of vaccination in previously infected children and under what circumstances those benefits accrue. We present a novel methodological framework for estimating the potential benefits of COVID-19 vaccination in previously infected children aged five to eleven, accounting for waning. We apply this framework to the UK context and for two adverse outcomes: hospitalisation related to SARS-CoV-2 infection and Long Covid. We show that the most important drivers of benefit are: the degree of protection provided by previous infection; the protection provided by vaccination; the time since previous infection; and future attack rates. Vaccination can be very beneficial for previously infected children if future attack rates are high and several months have elapsed since the previous major wave in this group. Benefits are generally larger for Long Covid than hospitalisation, because Long Covid is both more common than hospitalisation and previous infection offers less protection against it. Our framework provides a structure for policy makers to explore the additional benefit of vaccination across a range of adverse outcomes and different parameter assumptions. It can be easily updated as new evidence emerges.

## 1. Introduction

The Pfizer-BioNTech paediatric COVID-19 vaccine for five- to eleven-year-olds has been approved in the US, EU and UK since late 2021.

Before Omicron emerged at the end of 2021, when most children had not yet been infected and the vaccine displayed excellent efficacy [1,2,3], roll-out in the US [4], Israel and European countries was motivated by a desire to prevent adverse outcomes in children infected with COVID-19 for the first time. The Omicron SARS-CoV-2 variant proved both more transmissible and more immune evasive, with more rapid waning of protection from previous infection or vaccination [2,5,6,7,8,9,10,11,12,13,14,15], and a much higher incidence of reinfections [16]. Protection against new infection and subsequent hospitalisation has been shown to be highest in children who had both vaccination and previous infection [3,9,10]. Protection against adverse outcomes can be very high shortly after infection, but as protection wanes over the following months and years [15], adding further protection through vaccination might provide significant benefit. The question is then: how much benefit and when would vaccination (with respect to previous exposure) maximise any benefit?

In England, three large Omicron waves from January to July 2022 resulted in a very high number of infections in children [17]. Population antibody surveys estimate that 82% of primary school children had antibodies to SARS-CoV-2 in March 2022 [18], and almost all of these antibodies would have been as a result of exposure to SARS-CoV-2 and not vaccination (as general roll-out did not begin until April 2022). Not all children seroconvert and some will serorevert over time [19]. As such, given continued infections since then (including a large July 2022 wave in five- to eleven-year-olds), 82% is likely a significant underestimate as of February 2023. Estimates of incidence from the UK Office for National Statistics (ONS) Coronavirus (COVID-19) Infection Survey [20] show that the highest number of new infections over the whole of the pandemic is in children aged two to eleven [up to November 2022] and cumulative incidence in two- to eleven-year-olds was 98% over the first two Omicron waves (December 2021 to June 2022) [21]. 

While most high-income countries offered COVID-19 vaccination to children aged five to eleven [22,23,24,25], uptake remains low. In England, vaccines were offered to five- to eleven-year-olds from April 2022, but COVID-19 vaccination for five- to eleven-year-olds with no underlying health conditions will end from June 2023 [26]. As of April 2023, only 10% of five- to eleven-year-olds have received at least one dose [27]. The vaccination coverage in this age group will decrease every year as under-fives (who have no access to the vaccine in the UK) and vaccinated 11-year-olds grow older, entering and leaving this age group, respectively. As protection from previous infection wanes, should this decision to stop primary vaccination for five- to eleven-year-olds be revisited?

A revised assessment of vaccine benefit for five- to eleven-year-olds is required to reflect the current situation, namely the *added benefit* of a vaccine dose over the protection provided as a consequence of previous infection. The wording of “a vaccine dose” is deliberate: while the efficacy of a vaccine dose wanes relatively rapidly in children in the context of the Omicron variant (particularly against infection), protection can be increased again through subsequent doses, whether that is a second dose [7,12,13,28] or a booster dose [9,13]. We must consider waning from vaccination and infection in combination with population infection dynamics to determine not just whether to vaccinate but also when.

With a soup of new, more immune-evasive Omicron variants continuing to emerge [29,30] and national prevalence fluctuating between 1.5–8% since January 2022 [17], it seems likely that new Omicron variants and waning immunity will continue to keep prevalence above 1.5% in the near future [31,32]. 

Whilst protection against serious outcomes such as hospitalisation and death persists for longer than protection against infection, avoidance of reinfection remains important in view of emerging evidence on the association between the number of reinfections and the burden of acute and post-acute COVID-19 sequelae (albeit most relevant studies pertain only to adults) [33] and in view of the ONS Infection Survey data showing no evidence of reduction in reported Long Covid in children upon reinfection compared to first infection (albeit in a relatively small sample) [34].

Current mathematical models for estimating vaccine benefit often do not take waning into account (see Box 1). Waning is complicated by the fact that the extent and duration of immunity against Omicron reinfection depends on which variant, or sub-variant, caused the initial infection and when that infection occurred [35]. These uncertainties make the use of already complex SEIR or agent-based models of infections and outcomes even more challenging. In this paper, we present a simple but flexible framework for estimating the likely additional benefit of a vaccine dose in children aged five to eleven with previous infection, accounting for waning of immunity. We illustrate the framework in the UK context using current best evidence for averting hospital admissions related to SARS-CoV-2 infection and new cases of Long Covid.

Box 1Existing models for risk/benefit of vaccination against SARS-CoV-2 in 5–11-year-olds.    When the US Centers for Disease Control and Prevention (CDC) carried out its risk-benefit assessment in November 2021 [4], it based its calculations on the projected different levels of incidence into the future and estimated prevented new infections, hospitalisations and cases of Multisystem Inflammatory Syndrome in Children (MIS-C). The CDC assessment acknowledged that at that point, 38% of the age group had had at least one infection but considered that infection would not provide 100% protection and that it waned with time. The CDC (necessarily) used efficacy estimates for the vaccine against Delta and assumed no new variants. Borchering et al. [36] reported on an ensemble of nine models (mostly SEIR or agent-based), assessing the impact of vaccination of five- to eleven-year-olds in the US for a more transmissible variant than Delta and finding that vaccination of five- to eleven-year-olds would provide direct benefit. Four out of nine of the models did not consider waning after infection, four assumed a fixed amount of waning after a number of months and one used an exponential waning model. Only two incorporated waning after vaccination on the timescale of the projections (6 months). Waning against new infection or more severe outcomes was not explicitly differentiated.    Hawkes and Good [37] modelled the roll-out of the five- to eleven-year-old vaccine in Canada using a deterministic susceptible–infected–recovered (SIR) model and found modest clinical benefit to children at essentially zero risk. However, their model assumed that immunity from either vaccination or infection did not wane and their SIR models do not allow for reinfection.    Keeling and Moore have also modelled the impact of vaccination in five- to eleven-year-olds in England [38]. Their most recent reported run of the model was in November 2021 (prior to Omicron), with the results informing the decision making of the UK approval body, the Joint Committee for Vaccination and Immunisation (JCVI). They showed modest clinical benefit across a range of uptake estimates, assuming that roll-out would begin in March 2022. Protection from the vaccine or previous infection is assumed to be constant (no waning) [39,40], meaning little benefit of vaccination is *intrinsically* possible after large waves of infection because children are assumed to already be protected.    The JCVI had not approved the five to eleven vaccine for this age group in February 2022, by which time a large proportion of five- to eleven-year-olds had been infected in the January 2022 wave. Thus, for its February 2022 deliberations, the JCVI commissioned mathematical models of two future wave scenarios: a more serious or less serious variant, with different attack rates (33% and 27%, respectively) and hospitalisation rates (0.064% and 0.023%, respectively). The models additionally assumed that 80–90% of children had been previously infected and that previous infection provided 50–70% additional protection against disease outcomes. No waning was considered from either vaccination or past infection, and vaccine effectiveness was considered the same whether children had been previously infected or not (i.e., no benefit was ascribed to hybrid immunity) [41]. On the basis of the results of these models, in February 2022 the JCVI assessed that the benefit of the vaccine did outweigh the risks, but that any benefit was marginal, with between 78 (less severe variant) and 450 hospital admissions (more severe variant) prevented if all children without an underlying health condition were vaccinated [41].

## 2. Methods

In the below analysis, we assume all children have been previously infected (note that this underestimates the vaccine benefit, since the benefit will be larger in the absence of previous infection). The timeline that frames our analysis is shown in Figure 1. We also assume, consistent with the evidence, that vaccination after infection can never reduce protection against adverse outcomes from subsequent reinfection. Notation used is summarised in Table 1.

### 2.1. Quantifying the Additional Benefit of Vaccination for an Individual Previously Infected Child

At time t+s since infection, the probability of an unvaccinated child experiencing an adverse outcome of type A if reinfected can be expressed as:(1)$$r_{A}\left(1-X_{A}\left(t+s\right)\right)$$
where $r_{A}$ is the probability of adverse outcome, A, in an unvaccinated child infected for the first time and $X_{A}$ is the degree of protection afforded by previous infection at time t+s. Protection ($X_{A}\left(s+t\right))$ can range from 0 (no protection) to 1 (complete protection).

If a child is vaccinated at time *T_V_* (i.e., $T_{0}$ + *s*
−4 weeks = $T_{1}-4$ weeks; see Figure 1), then the probability of that child experiencing an adverse outcome of type A if (re)infected at time t can be expressed as: (2)$$r_{A}\left(1-Y_{A}\left(t\right)\right)\leq r_{A}\left(1-X_{A}\left(t+s\right)\right)$$
where $Y_{A}$ is the degree of protection afforded by vaccination against an adverse outcome of type A at time t+4 weeks after vaccination in a previously infected child. Again, protection can range from 0 (no protection) to 1 (complete protection) but is (consistent with evidence) assumed to be at least as high as $X_{A}\left(t+s\right)$. The key (reasonable) assumption here is that protection of vaccination following previous infection **only** depends on time since vaccination and does not depend on the interval between previous infection and vaccination. Note we use four weeks to allow for maximal vaccine efficacy, following Lin et al. [7].

Therefore, the additional benefit (reduction in adverse event probability) offered by vaccination to a child infected *t + s* months ago can be expressed as:(3)$$r_{A}\left(1-X_{A}\left(s+t\right)\right)-r_{A}\left(1-Y_{A}\left(t\right)\right)=r_{A}\left(Y_{A}\left(t\right)-X_{A}\left(t+s\right)\right)$$

### 2.2. Incorporating Waning

For both vaccination and infection, efficacy wanes over time. The functional form of waning is currently uncertain, but Lin et al. [7] provide tables for waning efficacy of vaccination and previous infection against reinfection and waning efficacy of vaccination against hospitalisation (not disaggregated by previous infection status) in children aged five to eleven. While Lin et al. show good vaccine effectiveness by two weeks, maximal efficacy is at four weeks, after which waning begins. For both protection against reinfection and hospitalisation upon reinfection, waning from week four post-vaccination can be approximated by a linear relationship over a time frame of a few months (up to the end of the available data). Lin et al. also provide tables of effectiveness of previous infection alone against reinfection and hospitalisation upon reinfection, which we have plotted in Figure 2 [7]. While the waning over time is not linear, as a first approximation and on the timescale of several months, it is adequate for our use case. Waning is clearly much faster for protection against reinfection than it is for protection from serious illness as measured by hospitalisation upon reinfection. We assume that we can also use a linear approximation for waning following vaccination with previous infection.

If c and d represent the rate of waning following infection alone and vaccination after infection, respectively, then a linear relationship implies the following:(4)$$Y_{A}\left(t\right)=\max(0,\left(Y_{A}\left(0\right)-dt\right)\;\mathrm{and}\;X_{A}\left(t\right)=\max\left(0,X_{A}\left(s\right)-ct\right)=\max\left(0,X_{A}\left(0\right)-cs-ct\right)$$

It is reasonable to suppose that the rate of waning from vaccination following previous infection *cannot be faster than* waning from previous infection alone. So, a *conservative* estimate of additional vaccine benefit is to assume that the rate of waning from vaccination following previous infection is equal to that of infection alone, i.e., d=c in Equation (4).

Substituting Equations from (4) into Equation (3) and setting d=c then gives the following additional benefit to a child in reducing the probability of an adverse event of type *A*:
(5a)rAYA(0)−ct−XA(0)+ct+cs=rAYA(0)−XA(0)+cs,YA(t)>0andXA(s+t)>0(5b)rAYA(0)−ct,YA(t)>0andXA(s+t)=0(5c)0,YA(t)=0andXA(s+t)=0

Equation parts (5b) and (5c) simply say that the benefit is just the full vaccine benefit if protection from previous infection has waned to zero (5b), or the benefit is zero if we are sufficiently far in the future for the benefit from both infection and vaccination to have waned to zero (5c). 

### 2.3. Notation

As long as Equation (5a) holds, the *time since vaccination* does not make a difference to the *additional* benefit from vaccination for children previously infected. Instead, the benefit depends only on the difference between maximum protection of vaccination after prior infection ($Y_{A}\left(0\right)$), maximum protection after infection ($X_{A}\left(0\right)$) and time between last infection and 4 weeks after vaccination, s. For severe outcomes from reinfection, protection from previous infection is likely to be non-zero over long time periods (so Equation (5a) holds). Protection against reinfection will wane more quickly, but we note that since the benefit provided by Equation (5b) is strictly greater than that provided by (5a), considering only Equation (5a) will provide a lower bound of benefit even once protection from previous infection has dropped to zero (for long intervals *s + t*).

### 2.4. Estimating Additional Benefit for the Six Months Post-Vaccination following Previous Infection with Linear-Approximated Waning

We consider a timescale of six months for vaccine benefit as a plausible minimum interval between vaccine boosters and/or before the pandemic context might change from a new variant or an updated vaccine (so Equation (5c) does not apply). We consider a maximum time window of 15 months between infection and 4 weeks after vaccination (*s*). 

Over the whole population, the time since previous infection, *s*, varies, but given the linearity in Equation (5a) we can simply take expectations such that the expected reduction in the number of adverse events of type A for attack rate p is given by:(6)$$pN_{A}\left(Y_{A}\left(0\right)-E(X_{A}\left(s\right)\right)=pN_{A}\left(Y_{A}\left(0\right)-X_{A}\left(0\right)+cE\left(s\right)\right)$$

We also need to consider the possibility of adverse events from the vaccine. Thus, assuming all children were vaccinated, the final expected reduction in the number of adverse events of type A is given by:(7)$$pN_{A}\left(Y_{A}\left(0\right)-X_{A}\left(0\right)+cE\left(s\right)\right)-N_{VA}$$
where *N_VA_* is of adverse events of type *A* due to the vaccine across the whole population of *N* children, if all were vaccinated. 

The key is that as long as we have a reasonable estimate for $N_{A}$ (number of adverse events following *first* infection if *all* children were infected) and reasonable estimates for maximal vaccine effectiveness for children with previous infection, $Y_{A}\left(0\right)$, then we can quantify the additional benefit of vaccination across a range of estimates for effectiveness of protection of infection after several weeks and months $\left(X_{A}\left(0\right)-cE\left(s\right)\right)$ and the proportion of children infected in the future, *p*. This framework can be applied to the child population as a whole, or to sub-cohorts (such as children with and without underlying health conditions).

### 2.5. Parameterising the Framework and Depicting Plausible Benefit

Plausible estimates drawn from the literature for the relevant parameters are given in the following sections. The UK Office for National Statistics (ONS) Coronavirus (COVID-19) Infection Survey has released estimates of the cumulative incidence by age group and by variant up to November 2022. Among two- to eleven-year-olds, it estimates a cumulative incidence of 48% over the seven-month Delta period; 60% over the three-month Omicron BA.1 period and 38% over the four-month Omicron BA.2 wave (98% over seven months) [21]. Autumn 2022 saw much smaller waves in two- to eleven-year-olds, but nonetheless ONS reports a cumulative incidence of 26% in two- to eleven-year-olds from June to November 2022 [21]. We thus consider a full range of future attack rates over six months from 0% to 100%.

By using a plausible estimate for maximum effectiveness of the vaccine following previous infection ($Y_{A}\left(0\right))$ and previous infection alone ($X_{A}\left(0\right))$ and using a plausible estimate for the waning rate, c, we can assess the possible additional benefit from vaccination across possible future attack rates and a range of plausible intervals, *E*(*s*), between infection and 4 weeks after vaccination from 4 months (the minimum) to 15 months using contour plots. 

For a given estimate of average time from last infection to four weeks post-vaccination, *E*(*s*), we can also estimate maximum and minimum additional benefit of vaccination for maximum and minimum parameter estimates, and plot the plausible range across all values of future attack rates, *p*. Essentially this depicts a cross section of the contour plot incorporating parameter uncertainty.

We now illustrate this framework with two important adverse outcomes. Firstly, the most commonly considered adverse outcome where there is also excellent real-world data: hospital admissions related to a SARS-CoV-2 infection, stratified by the presence of an underlying health condition. Secondly, we consider Long Covid, where estimates are far more uncertain, but the number of children affected is potentially much larger. Other models of vaccine benefit have typically not taken Long Covid into account due to uncertainties involved, so it provides an opportunity for demonstrating the utility of our simple framework.

We note that we do not explicitly consider new cases of paediatric inflammatory multisystem syndrome temporally associated with COVID-19 (PIMS-TS or MIS-C in the US) within the hospitalisation adverse outcome. PMS-TS is a severe adverse outcome of SARS-CoV-2 infection in children occurring typically 4 to 6 weeks post-infection [42]. While there is evidence that vaccination reduces rates of PIMS-TS [6,43], the extent to which previous infection or hybrid immunity protect against PIMS-TS is very uncertain. Thankfully, rates of PIMS-TS are also much lower than hospitalisation rates. Thus, in this paper, we use overall hospitalisations related to SARS-CoV-2 infection as an adverse outcome of interest, which will include PIMS-TS admissions. As more evidence emerges on PIMS-TS following reinfection (with or without vaccination), the framework could also be applied explicitly to PIMS-TS as an outcome.

## 3. Results

### 3.1. Hospital Admissions Averted

Table 2 gives the parameters used for applying the model to hospital admissions averted for children with and without an underlying health condition (UHC). Further detail on the parameterisation is given in the Appendix A. We note that we explicitly exclude incidental hospitalisations with SARS-CoV-2 infection from our parameter estimates, using new analysis from Wilde et al. (under review) [44]. 

The benefits across a range of possible intervals between previous infection and vaccination and all possible 6-month future attack rates are shown as contour plots in Figure 3 for children with (right panel) and without underlying health conditions (left panel) for a mid-range estimate of efficacy, assuming $Y_{A}\left(0\right)=100\%$, $X_{A}\left(0\right)=99.5\%$ and a waning rate c=1.3% points/month (see Table 2).

The largest additional benefit is when a large proportion of children are infected (high attack rate) *and* the last significant wave of infection in children was a long time ago (over a year) (right hand top corner). Conversely, there is little added benefit when future attack rates are low *or* infection is relatively recent. This is because protection against hospitalisation from new infection is very high (99.5%) shortly after previous infection and there is little room for improvement from vaccination even at high attack rates.

Children with underlying health conditions are more vulnerable to severe disease (as shown by Wilde et al. (under review) [44]) and the estimated number of hospitalisations averted with vaccination is slightly higher than in children with none of the specified types of these conditions [26], but at the cost of far fewer vaccinations required (0.7 million) [7].

Many children aged five to eleven years in England are now approximately a year out from their previous infection (two very large waves in January and March 2022 and a smaller wave in July 2022 [17]). Whilst previous infection is very protective against hospitalisation on reinfection in the months immediately following first infection, this protection does wane with time. Lin et al. [7] estimate an efficacy of 86.7% at 10 months post-infection. Our central, minimum and maximum waning rates (see Table 2) provide a range of efficacy at 1 year of 84% (79–92%) from previous infection alone. Vaccinating at a year would (under our framework) restore efficacy to 100% immediately after vaccination (regardless of how much protection from previous infection has waned), which then wanes slowly over the following 6 months. As we are assuming waning happens at the same rate for vaccination following infection and from infection alone (a conservative estimate), the added benefit of vaccination over infection alone is simply the difference in efficacy immediately after vaccination. Vaccination an average of a year after previous infection thus adds between 8% (=100–92%) and 21% (=100–79%) protection. An illustrative range of potential benefit in terms of averted hospitalisations on reinfection for different 6-month future attack rates is shown in Table 3. 

### 3.2. Cases of Long Covid Averted

Given the greater uncertainty in Long Covid parameters, we illustrate the range of benefit expected at a given average interval between infection and 4 weeks after vaccination, namely one year (E(s)=12). We thus wish to parameterise Equation (7) at *E*(*s*) = 12 for cases averted: $pN_{A}\left(Y_{A}\left(0\right)-X_{A}\left(12\right)\right)-N_{VA}$.

Table 4 gives the parameters used for applying the model to cases of Long Covid averted. Further detail on the parameterisation is given in Appendix A.

Figure 4 illustrates the expected number of additional cases of Long Covid over six months that could be averted by vaccination assuming an average time since infection of one year across the range of plausible benefit in children for a range of the proportion of children reinfected over that time period (attack rates). Note that for this example, we need to use estimates of protection from infection alone a year later ($X_{A}\left(12\right)$) and from vaccination 4 weeks after administration ($Y_{A}\left(0\right)$). Specific estimates at given future attack rates are also provided in Table 3 (essentially cross sections through Figure 4).

Firstly, while the pattern of increasing benefit is the same as for hospitalisations, the potential scale of cases averted is far higher (potentially 10,000–75,000 Long Covid cases averted in a medium 6-month wave with attack rates of 50–60%, if average time between previous infection and vaccination is a year). Secondly, the plausible range of benefit is very wide, reflecting the large uncertainty in the evidence around the protection that vaccination and/or previous infection provide in preventing both infection and Long Covid once (re)infected. 

## 4. Discussion

The benefit of vaccination in preventing adverse outcomes on new infection for children previously infected will be lower than its benefit for infection-naïve children [12]. Extrapolating the UK’s experience [21], it is likely that almost all children aged five to eleven in many countries will have experienced at least one SARS-CoV-2 infection. Thus, given that vaccine uptake in this age group remains much lower than for adults, it is important to try to quantify what the additional benefit of vaccination is in previously infected children, especially as primary vaccination in healthy children is due to be withdrawn in the UK from the summer of 2023. In this paper, we have suggested a new framework for understanding the potential value of COVID-19 vaccination, incorporating waning and providing a robust and efficient method for illustrating both the range and the magnitude of possible benefit and the extent of uncertainty in those estimates given a set of modifiable parameters (e.g., duration since prior exposure, waning of immunity from previous infection and vaccination, vaccine side effects).

### 4.1. Benefit

Applying this framework to the case of five- to eleven-year-olds in England for a future moderate-sized 6-month attack rate of about 50% [21], assuming an average of a year since previous infection, illustrates that there might plausibly be a relatively modest but important benefit with regard to hospitalisation risk from vaccinating all children without an underlying health condition of around 330 averted hospitalisations in a population of 4 million. Thus, the size of anticipated benefit on SARS-CoV-2-related hospitalisations even a year after previous infection is certainly not negligible but is considerably lower than the over 2000 hospitalisations expected if 50% of children were infected for the first time. There would be a similarly sized benefit from vaccinating all of the much lower number (~700,000) of children with any evidence of an underlying health condition placing them at increased risk of severe disease [26]. The relatively modest benefit is because previous infection alone is very protective against new hospitalisation and that protection wanes relatively slowly. That said, as the time since previous infection increases, there is benefit to a “top-up” vaccination and this will only increase further as more time elapses without new significant waves of infection in this age group. Indeed, Figure 3 illustrates that the benefit of vaccination following previous infection is much higher now in the UK than it would have been in spring 2022 (when vaccination was first offered) since many more five to eleven year old children are now a year or more out from their most recent infection [17,21]. 

The potential benefit In preventing Long Covid is potentially much greater (tens of thousands of Long Covid cases averted) but the true figure is much harder to ascertain due to the large uncertainty in the estimates. Given the potential scale of benefit, this highlights the urgency of understanding better the protection from previous infection and hybrid immunity against Long Covid. We note that even if most cases of Long Covid in children resolve within a few months [54], those months still represent significant disruption to a child’s education and life more broadly.

### 4.2. Strengths and Limitations

The key strength of this study is that the underlying modelling framework is flexible, allowing for a range of future scenarios (e.g., one can lower or increase range of efficacy to model new variants, more waning or better vaccines). Moreover, the framework incorporates adverse effects from the vaccines (Equation (7)). Finally, the framework can be easily extended to adults or any other sub-population of interest to explore the benefits of further vaccine doses given previous infection or updated with more accurate parameter estimates as data become available, as long as the assumptions made are reasonable in that population. 

However, there are four key limitations of the current framework. First, several assumptions were made in the model of benefit. Critically, we assumed that waning of protection occurs in a linear fashion based on available data [7], but this assumption is very likely to fail for longer time scales. Second, we assumed that the rate of waning protection from prior infection versus vaccination following infection were identical, which is also likely to be false (although, if anything, waning of hybrid immunity will be slower than from infection alone, thus this assumption would underestimate the benefit of vaccination post-infection). Third, the parameter estimates around waning and protection offered by vaccination or infection against Long Covid are still very uncertain. Fourth, the framework itself only considers the direct benefits of individual protection.

There are a number of indirect benefits as well which we have not considered. For example, a vaccinated cohort is less conducive to community transmission/spread, since hybrid immunity has been consistently shown to be higher and longer lasting than immunity from infection alone [7,10,51]. This would not only have knock-on benefits for other children, thereby further reducing risk of infection and adverse events, but would also break transmission chains which could result in older or otherwise vulnerable individuals contracting COVID-19, for whom there is a much higher risk of severe outcomes [55]. Furthermore, there are additional indirect benefits to the vaccinated individuals that have not been considered, including shorter or less severe Long Covid if it occurs [56] and reduced school-related absenteeism [57] as a consequence of reduced infection and Long Covid rates, that can have longer term health-related and socio-economic impacts [58]. 

On the other hand, we have also not considered potential indirect negative consequences of vaccination, such as missing school due to short-lived vaccine side effects or any impact on uptake of other childhood vaccines [41]. Finally, certain logistical assumptions have been made, e.g., assuming all vaccination happens instantaneously at a given timepoint or that all children will be vaccinated, which is not realistic. In light of these limitations, it is worth emphasising that this framework is not meant to replace a formal health economic analysis, but rather to move the debate about childhood vaccination forward acknowledging both widespread previous infection and waning protection from infection and vaccination.

### 4.3. Policy Implications

Decisions on SARS-CoV-2 vaccination in many countries were originally made as new vaccines became available and new variants emerged. Over the last year, most countries have now been moving to an offer of regular boosters (once or twice a year) for more vulnerable adults (e.g., UK, France, Italy) or more rarely, for all adults (e.g., Australia) or both adults and children (e.g., US, Singapore). 

Many countries are not just excluding children from their booster programmes, but are withdrawing the offer of a primary course of vaccines for five- to eleven-year-olds with no underlying health conditions (e.g., UK [26], Sweden [59], Denmark [60]). The rationale given is that most children have a mild course of illness and are protected by previous infection. However, this does not take into account waning protection from previous infection. This combined with the ongoing high levels of prevalence in the population will lead to more cases of severe illness and of Long Covid.

Our framework highlights that vaccination campaigns immediately following a large wave of infection are inefficient in terms of added benefit. Ideally, vaccination campaigns would be timed several months to a year after large infection waves and before a new wave. If SARS-CoV-2 becomes more seasonal, similar to influenza or RSV, a vaccination campaign could be planned annually in the late summer before school terms start and provide additional good protection through the winter season. Once (if) a clear seasonal pattern emerges, vaccination decisions could be revisited by national bodies. 

However, if SARS-CoV-2 retains high levels of infections year round with frequent, smaller waves of infection [61] or if new, big waves can arise at any time of year (with a new variant), then planning a vaccination campaign to maximise benefit is much harder. Possible strategies might be to institute randomized population testing (similar to the now-discontinued ONS infection survey [17]) to keep track of levels of infection and/or antibodies to SARS-CoV-2 in children and plan new vaccination campaigns once the average time since previous infection is over a certain value (e.g., a year). Such a strategy could be tailored to several age groups simultaneously and take into account immune escape properties of currently circulating variants. 

An alternative strategy might be to provide vaccination on demand, but with the advice that children do not receive a vaccine dose within six to nine months of infection. Of course, that is much harder to implement now that tests are not freely available, far fewer people test and children are frequently asymptomatic upon infection, although receiving vaccines shortly after infection is inefficient rather than harmful. The difficulty and potential costs of incorporating timing since previous infection could also support a decision not to prioritise vaccination in this age group, instead focusing on children with underlying health conditions or those age groups more likely to be facing their first infection (under-fives). In such a strategy, care would need to be taken to regularly review the criteria for more vulnerable groups based on the latest evidence.

While the achieved benefit from vaccination following infection is relatively modest in terms of hospitalization in children with no underlying health conditions, Long Covid remains a key adverse outcome, with large uncertainties in incidence and degree of protection from previous infection, vaccination or both. The ONS infection survey reported that 0.6% of children experienced ongoing symptoms 20 weeks after their *second* infection [34], considerably lower than current estimates (see Table 4 and Appendix A) for persisting symptoms at two to three months (shorter timeframe than 20 weeks and from mostly first infections). Even at that lower estimate, 0.6% still translates to 12,000 children if 50% of the 4 million children with no underlying conditions were re-infected in a future wave, without vaccination.

Finally, we note that this framework, if adapted for adults, could be used to inform future booster campaigns for adults, whether for more vulnerable populations or all adults.

### 4.4. Future Research

More research is clearly required to evaluate the societal, health and economic cases for vaccination in the five to eleven-year-old population in the presence of widespread previous infection. The use of this framework highlighted where there are several key gaps in knowledge, which unless addressed will continue to limit the ability of policy makers and practitioners to fully understand the value of vaccination in this sub-population and others. For instance, more evidence is needed on how long protection from previous infection lasts, how long hybrid immunity lasts, how immunity wanes after several infections (with and without vaccination) and how this varies based on the type of vaccine or variant of previous infections.

## 5. Conclusions

We presented a framework for visualising the additional benefit of vaccination in children given high levels of previous infection. The framework provides a way to estimate plausible ranges of benefit as well as identify where important research gaps remain. The framework allows for a synthesis of real-world evidence, modelling and projected scenarios to inform policy discussion. While our example is centred on the UK context, the basic framework is applicable to any country or region with appropriately defined parameters. 

In essence, we illustrate that there is robust evidence for net benefit from continued vaccination of the five- to eleven-year-old cohort, even after previous infection with SARS-CoV-2, where the scale of benefit depends most strongly on the future attack rate and the time since last infection. In essence, the question that this framework exposes as being critical to an evidence-based policy is: at what point since previous infection would it be beneficial to *add* the immunity from the vaccine, given that we know that both vaccine-related and infection-related immunity wane?

As we learn more about the risks of adverse outcomes from third or more infections and the efficacy of vaccination in those with many previous infections, the parameters can be updated accordingly without changing the framework.

## Figures and Tables

**Figure 1 vaccines-11-00988-f001:**
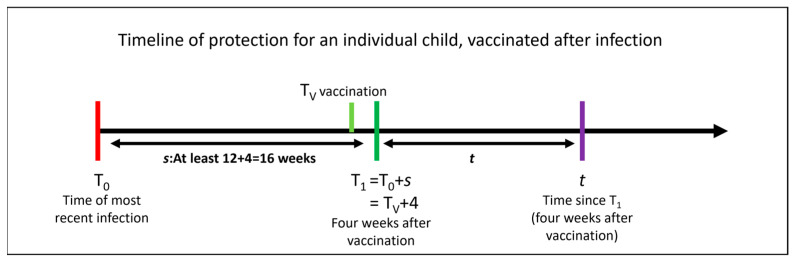
Timeline showing sequence of events for vaccination following prior infection. Note we use four weeks to allow for maximal vaccine efficacy, following Lin et al. [7].

**Figure 2 vaccines-11-00988-f002:**
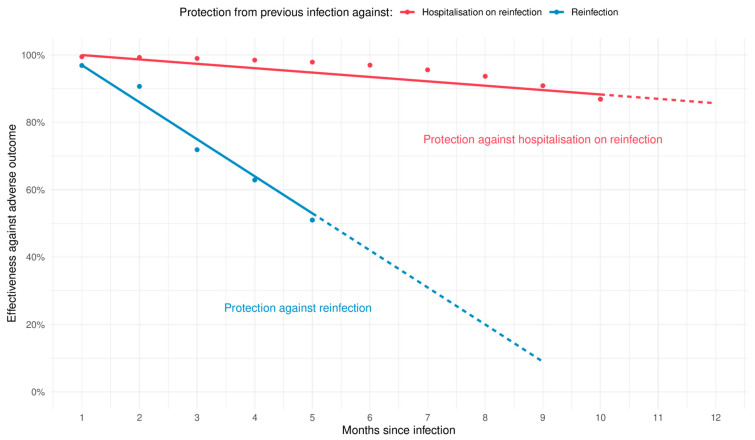
Charts showing the approximate linear relationship of waning of protection from previous infection (without vaccination) vs. reinfection (blue) and hospitalisation on reinfection (red). Data are taken from the supplemental material from Lin et al. [7]. Linear fits are shown as solid lines, with dashed lines showing the linear extrapolation.

**Figure 3 vaccines-11-00988-f003:**
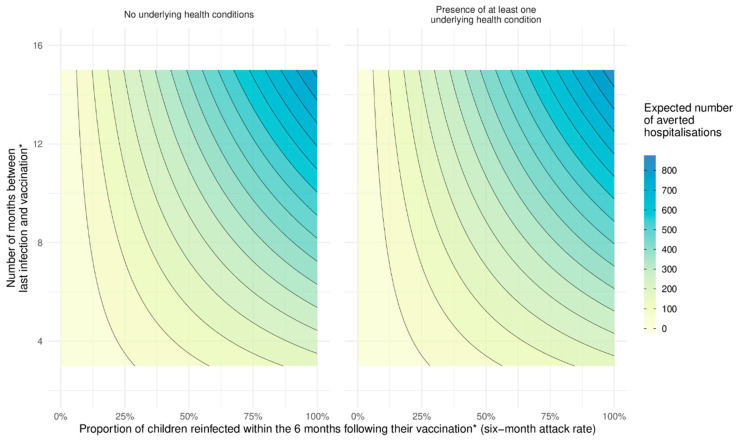
This figure shows the potential impact of vaccination amongst five to eleven -year old children in terms of the number of hospital admissions that can be averted across a range of months since previous infection to show the increasing impact of vaccination as both time since last infection and the six-month attack rate increase. * We define vaccination as the point in time at which 4 weeks have passed since the administration of a dose to allow for effectiveness to peak and begin to wane.

**Figure 4 vaccines-11-00988-f004:**
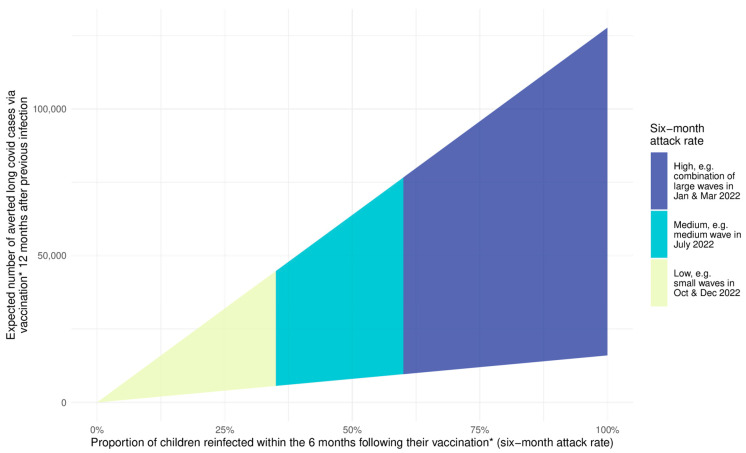
This figure shows the potential impact of vaccination amongst five- to eleven- year-old children in terms of the number of new Long Covid cases that can be averted under varying attack rate scenarios in the six months following vaccination*. We include the uncertainty in infection and vaccination protection to show an example range of outcomes for the case where 12 months separate previous infection and receipt of a vaccine dose before allowing for a further month to reach vaccination as defined above (Equation (7)). * We define vaccination as the point in time at which 4 weeks have passed since the administration of a dose to allow for effectiveness to peak and begin to wane [7].

**Table 1 vaccines-11-00988-t001:** Notation used to set up the new framework.

Variable	Description
A	Adverse outcome of type A due to infection
VA	Adverse outcome of type A due to vaccination
$T_{0}$	Time of a child’s most recent infection
$T_{V}$	Time of vaccination
$T_{1}=T_{V}+4$	Time of maximal vaccination efficacy in weeks (while Lin et al. show good effectiveness by two weeks, their reported maximal efficacy is at four weeks, after which waning begins [7])
s	Time between last infection and four weeks after vaccination ($T_{1}-T_{0}$), where this is at least 16 weeks (allowing 12 weeks between infection and vaccination, as per current UK guidance)
t	Time since four weeks after vaccination
$X_{A}\left(s+t\right),\;0\leq X_{A}\left(s+t\right)\leq 1$	The degree of protection afforded by previous infection against adverse outcome of type A at time s+t after infection in an unvaccinated child. Protection can range from 0 (no protection) to 1 (complete protection)
$Y_{A}\left(t\right),\;0\leq X_{A}\left(s+t\right)\leq Y_{A}\left(t\right)\leq 1$	The degree of protection afforded by vaccination *after previous infection* against adverse outcome of type A at time t after administration + four weeks (to allow for maximum efficacy). Protection can range from $X_{A}\left(s+t\right)$ (same as previous infection alone) to 1 (complete protection)
$r_{A}$	Probability of adverse event of type A for a child infected *for the first time*
$r_{VA}$	Probability of adverse event of type A due to the vaccine
$N_{VA}$	The number of adverse events of type A caused by the vaccine if the whole population of children were vaccinated
$N_{A}=r_{A}N$	Number of adverse events of type A across the whole population of N children, if all infected *for the first time*
$N_{VA}=r_{VA}N$	Number of adverse events of type A due to the vaccine across the whole population of N children, if all vaccinated
p, 0≤p≤1	The proportion of children infected (again) in the future, over some time period, where the proportion can range from 0 (no new infections) to 1 (all children reinfected)
c,d	The rate of waning of protection following infection or vaccination following infection, respectively

**Table 2 vaccines-11-00988-t002:** Main parameters used to quantify hospital admissions averted in children with and without an underlying health condition (UHC) aged 5–11.

Parameter for Children Aged 5 to 11	Value for Those with UHC	Value for Those with no UHC	Comment (Further Detail Provided in Appendix A)
Number in population resident in England	687,935	4,036,891	We use 2021 UK Office for National Statistics (ONS) estimates for overall population of 5–11-year-olds [45]. We derive a UHC rate of 14.6% based on the proportion of medical records for all of the five- to eleven-year-olds in the NHS Digital Trusted Research Environment where there was evidence of a health condition linked to greater vulnerability to severe disease with SARS-CoV-2 infection by the JCVI [26] as per the method described in Wilde et al. (under review) [44]. This rate is then applied to the overall ONS population estimate.
Proportion infected at least once by March 2022	82%		We use the UK Office for National Statistics (ONS) Schools Infection Survey antibody study for primary school children from early 2022 reporting that 82% of primary-school-age children had antibodies. The actual proportion infected is likely to be much higher (ONS infection survey estimates cumulative incidence of 123% by March 2022 [21]), so this is a conservative estimate. We assume the same proportion for children with UHC and those without UHC since we do not have a reliable way of estimating difference in infection likelihood between the two populations.
Number of hospital admissions directly associated with first ascertained SARS-CoV-2 infection July 2020 to end February 2022	3375	3465	We use the number reported in Wilde et al. (under review) [44] for first hospital admissions associated with first ascertained SARS-CoV-2 infection in children aged five to eleven and excluding admissions that are incidental to SARS-CoV-2 infection (see Wilde et al. for details), from July 2020 to end of February 2022 in England. NOTE: this will be an underestimate of all hospitalisations since it excludes individuals that had an admission prior to July 2020.
Number of hospital admissions expected *if whole population had been infected for the first time* ($N_{A}$)	4120	4230	Obtained by dividing the number observed by the proportion of children infected in each group as given in row 2 above, to three significant figures.
Maximum protection of previous infection against hospitalisation from reinfection $(Y_{A}\left(0\right)$)	99.5%		Lin et al. [7] give an estimate of 99.5% efficacy at 1 month post-infection against hospitalisation.
Maximum protection of vaccination in children with previous infection (at least 16 weeks after infection and 4 weeks after vaccination) $(X_{A}\left(0\right)$)	100%		Lin et al. [7] do not provide estimates for additional protection from vaccination but just say that it is higher than vaccination or infection alone. Bobrovitz et al. [9] give an efficacy of over 95% at both three and twelve months after two or three doses of vaccine following infection.
The waning rate of protection from either vaccination following infection or infection alone (c)	1.3 percentage points per month (minimum 0.6 and maximum 1.7)		Estimated using the data in the supplemental material for Lin et al. [7] for protection from previous Omicron infection only. See Appendix A for fits for maximum and minimum ranges and Figure 2 for the central estimate.
The number of hospitalisations due to vaccine adverse events *if all children were vaccinated* (*N_VA_*)	1	7	A systematic review of vaccination for 5–11-year-olds reported a myocarditis rate of 1.3–1.8/million vaccinations given [46]. Extrapolating this rate to the five to eleven year old population in UK (five million) and conservatively assuming all would be hospitalised gives a central estimate of eight hospital admissions due to the vaccine, which we split across UHC and non UHC children 1:7. We note that Watanabe et al. report no vaccine-caused deaths among 16.6 million injections [46].

**Table 3 vaccines-11-00988-t003:** Illustration of plausible range of benefit in terms of hospitalisations and new cases of Long Covid averted for vaccination an average of a year after previous infection for a range of attack rates in the 6 months following vaccination. All numbers are rounded to reduce impression of precision and with plausible minimum and maximum estimates of efficacy (see Table 2 and Table 4).

Attack Rate (Proportion of Children Reinfected in the 6 Months Post-Vaccination)	Estimated Number of Hospitalisations Averted on Reinfection if All 4 Million Children without Underlying Health Conditions (UHC) Were Vaccinated on Average a Year after Previous Infection	Estimated Number of New Cases of Long Covid Averted on Reinfection if All 4.7 Million Children Were Vaccinated on Average a Year after Previous Infection
5%	25 (10–35)	3500 (800–6500)
25%	160 (75–215)	18,000 (4000–32,000)
40%	265 (120–350)	29,000 (6500–51,000)
50% (ONS estimate an incidence rate of 45% in 2- to 11-year-olds July–December 2022)	330 (160–440)	36,500 (8000–64,000)
60%	400 (190–520)	43,500 (9500–76,500)
75%	500 (240–660)	54,500 (12,000–96,000)
95% (ONS estimate an incidence rate of 98% in 2- to 11-year-olds, Dec 2021 to June 2022)	640 (300–830)	69,000 (15,000–121,500)

**Table 4 vaccines-11-00988-t004:** Main parameters used to quantify Long Covid cases averted in children aged 5–11.

Parameter for Children Aged Five to Eleven	Value	Comment
Number in population resident in England	4,724,826	We use 2021 UK Office for National Statistics (ONS) estimates for overall population of five- to eleven-year-olds [45].
Incidence of Long Covid (ongoing symptoms lasting at least 3 months following first infection)	3.5%	A central estimate from recent studies and the American Academy of Pediatrics [47,48,49,50].
Number of children experiencing Long Covid if whole population had been infected for the first time ($N_{A}$)	165,000	Rounded to three significant figures.
(Infection only): Protection of previous infection against new Omicron infection at *one year* post-infection (representing $X_{A}\left(0\right)-12c$)	16–36%	Bobrovitz et al. [9] give an efficacy of 24.7% at 12 months with CI 16.4% to 35.5%.
Hybrid): Maximum protection against reinfection of vaccination in children following previous infection. We are interested in efficacy *just after vaccination* $Y_{A}\left(0\right)$ *)*	59–78%	Bobrovitz et al. [9] give an efficacy of 69.0% at 3 months with CI 58.9% to 77.5%, and we use this as a conservative estimate of maximum vaccine benefit following previous infection. Note also that Dowell et al. [51] show that antibodies to SARS-CoV-2 are increased greatly in children with vaccination on top of previous immunity (from Omicron).
(Infection only): Minimum and maximum protection of previous infection against Long Covid once reinfected at least one year later	0–40%	There is a great deal of uncertainty in the protection afforded by previous infection once reinfected. The UK ONS reports no significant difference in reporting Long Covid in 2- to 11-year-olds 20 weeks after reinfection vs. 20 weeks after first infection [34]. We choose a range of 0–40% as a plausible range.
(Hybrid): Maximum protection of vaccination following previous infection against Long Covid once reinfected	30–70%	There is a great deal of uncertainty in the protection afforded by hybrid immunity once reinfected. Protection from vaccination alone is thought to be somewhere between 15–50% [52]. A recent systematic review reported great uncertainty in the scale but likely definite benefit of vaccination in preventing Long Covid [53]. Assuming hybrid protection is better than vaccination alone, we chose a range of 30–70%.
Overall effectiveness of previous infection one year earlier against new Long Covid on reinfection	16–62%	Combining minimum and maximum ranges of respective protections above.
Overall effectiveness of vaccination after previous infection against new Long Covid on reinfection	71–93%	Combining minimum and maximum ranges of respective protections above.
The number of Long Covid cases due to vaccination if all children were vaccinated	0	There is no mechanism by which vaccination can cause Long Covid.

## Data Availability

Not applicable as this study uses only publicly available aggregate estimates to parameterise the model.

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
