# Peer review of "A Methodological Framework for Assessing the Benefit of SARS-CoV-2 Vaccination following Previous Infection: Case Study of Five- to Eleven-Year-Olds"

_vaccines, 2023, doi:10.3390/vaccines11050988_

Round 1
Reviewer 1 Report
This is an interesting and well written manuscript on a framework on the benefits of COVID-19 vaccination among children. My main concerns are regarding what seems to be discussion in the results section, and a rather generally shallow discussion. Indeed, the discussion could be expounded to exhaustively elucidate the key issues emerging from the study, how they relate to existing literature (and current practice) globally, and the contribution of the results to informing policy, practice and programmes in the UK and around the world.
Reviewer 2 Report
This manuscript, by Pagel, C et al., presents a methodological framework for assessing the benefit of SARS-CoV-2 vaccination following previous infection in five to eleven-year olds in UK.
The manuscript presented the methodology that predicted the net benefit from continued vaccination of the five to 11-year-old cohort, even after previous infection with SARS-CoV-2, where the scale of benefit depends mostly on the future attack rate and the time since last infection.
The strength of this study is that the underlying modelling framework is flexible, and can be adapted to analyze adults or other sub-population of interest to give a reasonable reference for policy making.
The authors spent a long portion of discussion discussing about the key limitations of their work, which is conscientious on their part, However, it’s a bit overdoing and some points are quite trivial. One important question I think, especially when the health care policy makers were to use this framework as one of their references, they need to decide when is the best time they should give the vaccine to the children. In another word, should a serum titer be assayed first, then vaccinate those who’s levels are the lowest first, or just blindly given to everybody regardless?
